# Mitigating Context Bias via Foreground-Background Separation and Pooling: A Causal Analysis and Robust Evaluation

## Abstract

Context bias refers to the association between the foreground objects and background during the object detection training process. Various methods have been proposed to minimize the context bias when applying the trained model to an unseen domain, known as domain adaptation for object detection (DAOD). But a principled approach to understand why the context bias occurs and how to remove it has been missing. In this work, we provide a causal view of the context bias in the network architecture as the possible source of this bias. We present an analytical framework that utilizes an explicit foreground mask during feature aggregation with the proposed pooling operation to separate foreground and background, which leads the trained model to detect objects in a more robust manner under different domains. We use the ground truth masks and also masks generated using Segment Anything Model (SAM) to showcase the performance of the different state-of-the-art network model architectures such as ALDI++, ResNet, EfficientNet and Vision Transformer. We also provide a benchmark designed to create an ultimate test for DAOD, using foregrounds in the presence of absolute random backgrounds, to statistically analyze the robustness of the intended trained models using 95% confidence. Through these experiments, our goal is to provide a principled approach for minimizing context bias under domain shift for object detection.

## 1 Introduction

Domain adaptation for object detection (DAOD) addresses performance degradation caused by domain shifts between training and testing datasets. Despite advances in object detector architectures (He et al., 2017; Khanam & Hussain, 2024; Ren et al., 2016; Lin et al., 2017b; Tan & Le, 2019; Liu et al., 2016) and training strategies (Luo et al., 2024; Wang et al., 2025; Gevorgyan, 2022; Shermaine et al., 2025; Zoph et al., 2020; Triantafyllidou et al., 2024), models still struggle to generalize to unseen domains (Weinstein et al., 2022; Kay et al., 2022; Weinstein et al., 2021). Manual annotation on new datasets is expensive and labor-intensive. To alleviate this, sim-to-real approaches leverage synthetic data for training and real-world evaluation (Triess et al., 2022; Horváth et al., 2022; Kainova, 2023). However, the performance gap—known as the sim-to-real gap—remains substantial, and no method has surpassed the "ORACLE" model trained on fully labeled target data (Kay et al., 2025). This is largely due to spurious factors and distributional perturbations learned during training (Wu et al., 2023; Xu et al., 2023a; LaBonte et al., 2024; Qin et al., 2024). DAOD algorithms mitigate domain shifts through feature alignment (Ganin & Lempitsky, 2015; Chen et al., 2018; 2021; Zhu et al., 2019; Guan et al., 2021), adversarial domain classifiers (Ganin et al., 2016; Xu et al., 2023a), knowledge distillation (Kay et al., 2025; Pham et al., 2022; Caron et al., 2021; Chen et al., 2017), and hybrid methods (Cao et al., 2023; Deng et al., 2021; Xue et al., 2023; Hoyer et al., 2023). While these techniques aim to extract target-domain priors from consistent foregrounds (FG), recent research shows that context bias between FG and background (BG) leads to substantial deterioration of DAOD performance (Son & Kusari, 2024).

Causal tools can provide a natural way to measure the effect of the association between the FG and BG. In particular, Structural Causal Models (SCMs) provide a principled framework for capturing associational, interventional, and counterfactual causality (Schölkopf et al., 2021; Kusner et al., 2017; Kilbertus et al., 2017; Zhang & Bareinboim, 2018; Karimi et al., 2020; Von Kügelgen et al.,

2022), which traditional explainable AI (xAI) methods are not equipped to handle (Lundberg & Lee, 2017; Chen & Guestrin, 2016; Martin et al., 2021; Martin & Mahoney, 2021). Ultimately, modeling interventions to find causal discovery supports the development of predictive models that are robust to distributional shifts commonly encountered in real-world settings but also enables that can be extended to support reasoning.

Based on this knowledge, several recent studies applied causal interventions to DAOD (Zhang et al., 2024; Huang et al., 2021; Resnick et al., 2021; Jiang et al., 2023; Lin et al., 2022; Li et al., 2023; Xu et al., 2023b; Chen et al., 2022; Li et al., 2022b; Singh et al., 2020), with varying interpretations of context. Huang et al. (2021) treated context as a causal factor of the image and performed intervention using joint representations and attention mechanisms. Jiang et al. (2023) used causal intervention to capture the frequent co-occurrence of pairs in images (visual causality) as well as to account for the presence of a challenging subgroup. Other studies addressed contrast and spatial distribution biases as confounding factors in adaptation (Lin et al., 2022), or identify and mitigated target discrimination bias in teacher-student frameworks using conditional causal intervention (Li et al., 2023). Spurious correlations were also targeted through frequency-domain augmentation and multi-view adversarial discriminators to isolate domain-invariant causal features (Xu et al., 2023b). Additionally, ambiguous boundaries were resolved by modeling a category-causality chain and intervening with saliency-based causal maps (Chen et al., 2022). The SIGMA framework (Li et al., 2022b) reformulated DAOD as a semantic-complete graph matching problem to address mismatched semantics and intra-class variance. While not a formal application of causal intervention (e.g., using do-calculus or SCMs), SIGMA adopted causal reasoning principles to reduce spurious correlations caused by missing semantics or noisy BGs. (Qin et al., 2024) applied propensity score weighting (via Fourier Fast Transform and K-clustering) to study spurious features in classification.

However, it remains uncertain whether undesirably learned FG-BG association can impact DAOD. Also, whether the model utilizes the impact of spurious factors is unexplored.

These facts raise a critical question: *can commonly used backbones in model architectures introduce artifacts that confound learning and contribute to domain shift?* Despite extensive research on spurious features analysis, their causal implications remain unexplored. We hypothesize that FG and BG features, being inherently independent, may form spurious associations during training, ultimately impairing generalization to novel domains.

To investigate this, we design experiments leveraging object detection tasks to examine the causal relationship between FG-BG associations and model performance. We apply causal intervention-based analysis using FG masks generated from ground truth and SAM (Kirillov et al., 2023) to isolate effects and draw causal inferences. We focus on multi-object detection because it provides a balanced complexity—more intricate than image classification, yet more manageable than semantic segmentation—making it suitable for causal analysis. Our findings offer insights for developing models with stronger domain generalization and out-of-distribution (OOD) robustness. Our observation and experimental outcomes can provide remarkable perspectives on DAOD community.

Our main contributions are as follows:

- We demonstrate that general backbones can introduce causal artifacts that negatively affect model performance for DAOD task.
- We utilize an explicit FG-BG separation strategy using Mask Pooling to remove the causal effects of FG-BG associations on object detection outcomes and showcase the robustness empirically in different source and target domains using multiple neural network models.
- We present a benchmark to evaluate model robustness for DAOD task by creating synthesized images with known FGs juxtaposed on random BGs.

Through these contributions, we aim to provide a novel perspective to the DAOD community for mitigation context bias.

## 2 PROBLEM DEFINITION

The left panel of Fig. 1 illustrates three causal cases in DAOD to investigate context bias. Based on PC algorithm (Glymour et al., 2019), we construct causal graphs. Case (1) ("F" → "Y") represents cause-effect, where predictions rely solely on the distribution of FG features. Case (2) reflects

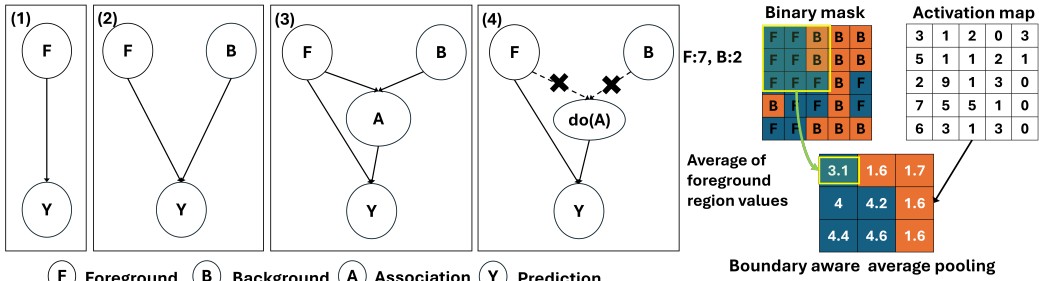

Figure 1: **Left panel**: (1) ideal case where the label "Y" is dependent only on the FG ("F"); (2) actual case where the FG ("F") and BG ("B") is associated in "A" and influences the prediction; (3) proposed model where the mask pooling removes association by making the pooling operation separately in FG and BG (Son & Kusari, 2024). **Right panel**: An example activation map with mask pooling is illustrated.

findings from a quantification study in object detection (Son & Kusari, 2024), modeled as a v-structure: "F" $\to$ "A" $\leftarrow$ "B"), where "A" encodes an implicit association between FG ("F") and BG ("B"), often unobservable. From Case (2), during training, Y is given by ground truth, "F" and "B" can be influenced each other so that it can cause "A" which affects prediction. Case (3) represents that the prediction target "Y" is influenced by both "F" and "A" ("F" $\to$ "Y" $\leftarrow$ "A") and "B" and "A" ("B" $\to$ "Y" $\leftarrow$ "A"). The corresponding joint distribution can be factorized as: $P(Y, A, F, B) = P(Y \mid F, A) P(A \mid F, B) P(F) P(B)$. Case (3) introduces causal intervention to analyze context bias by intervening on "A". This yields the interventional distribution: $P(Y, F \mid do(A))) = P(Y \mid F, do(A)))P(F)$ which simplifies to: $P(Y, F) \approx P(Y \mid F) \cdot P(F)$. From these identified estimands, we can define the training objective as an Empirical Risk Minimization (ERM) where mask pooling minimizes the interventional risk such that

$$\mathcal{R}_{do}(\theta) = \mathbb{E}_{(x,y)\sim\mathcal{D}}\Big[\ell\Big(y,\ \hat{f}_\theta\big(x;\ do(A)\big)\Big)\Big] = \mathbb{E}_{F,B,Y}\Big[\ell\Big(Y,\ \hat{f}_\theta(F, \tilde{A}(F, B))\Big)\Big]. \quad (1)$$

Because $\tilde{A}$ is a function of $F$ and $B$, $P(Y \mid do(A \leftarrow \tilde{A}))$ will marginalize FG-BG association. With respect to maximum likelihood, by the intervention, a model will be trained to increase the likelihood based on FG features and FG-BG associations because it follows anti-causal process verified mathematically (Kügelgen et al., 2020; Janzing & Schölkopf, 2015). However, it remains unclear whether suppressing or strengthening FG–BG associations benefits DAOD. Based on the counterfactual questions (e.g. what-if BG had been different?, what-if BG activation changes, still models are robust on it?), we design the experimental study to analyze FG-BG association impact on DAOD problem.

To investigate this, we compare models trained with and without FG masks. The use of masks enforces determinism in the variable "A" by explicitly separating FG from BG, thereby constraining the influence of spurious contextual features. Through causal intervention, we analyze the effectiveness of mitigating context bias and assess the robustness achieved by FG and BG separation.

## 3 METHODOLOGY

The preceding sections have revealed a hypothesis and the literature gap between DAOD and causality analysis in CNN-based deep learning algorithms. Based on the hypothesis, we design experiments to evaluate and quantify.

## 3.1 MASK POOLING SEPARATING FG-BG

$$P_{m,n} = \begin{cases} \dfrac{1}{n_F} \sum_{(i,j) \in K} F(i,j) \cdot x(i,j) & \text{if } n_F \geq n_B \\[2em] \dfrac{1}{n_B} \sum_{(i,j) \in K} (1 - F(i,j)) \cdot x(i,j) & \text{otherwise} \end{cases} \tag{2}$$

A FG mask is a binary mask of all FG objects. For example, Cityscapes Cordts et al. (2016) has 8 different FG classes. Given the masks, the pooling operates to compute average and voting based on the masks. Where $P_{m,n}$ is the pooled value, $K$ is the $3 \times 3$ kernel region, $x(i,j)$ is the pixel value at position $(i,j)$, and $F(i,j) \in \{0,1\}$ indicates whether $(i,j)$ is a FG pixel (1) or a BG pixel (0). $n_F$ is the number of FG pixels and $n_B$ is the number of BG pixels in the mask, respectively. The designed pooling separates each region using masks into FG and BG during training and inference and the average of each region is transferred respectively depending on voting systems. Fig. 1 and equation 2 illustrate how our proposed pooling layer works.

## 3.2 NEURAL NETWORK MODELS

We use 4 different models, including state-of-the-art ALDI++ and transformer-based ViTDet. ResNet-50 and EfficientNet-B0 are modified to evaluate the FG-BG separation via the pooling operation (See Table 1). "M-O" models are trained with ground truth binary foreground masks. "M" models are trained with SAM masks with morphological modification such as erosion and dilation.

Table 1: Comparison of Backbone Architectures and Modifications

| Model | Backbone | Mask Pooling Location | Notes |
|-------|----------|-----------------------|-------|
| ResNet-50 (He et al., 2016) | Single max pooling after stem | Replaced max pooling after first block (ResM) | Pretrained with COCO (Lin et al. (2014)). Integrated with FPN (Lin et al., 2017a) |
| EfficientNet-B0 (Tan & Le, 2019) | No explicit pooling except global average pooling | Added after stem layer (EffM) | Pretrained with ImageNet-1K. Global avg pooling unchanged; Integrated with FPN |
| ALDI++ (Kay et al., 2025) | ResNet-50 FPN backbone | | Pretrained with COCO. State-of-the-art DAOD model for comparison |
| ViTDet (Li et al., 2022c) | ViT backbone. No explicit pooling | | Pretrained with COCO. Integrated with FPN |

**Model abbreviations**:

*ALDI++*: Resnet-50 FPN with ALDI++; *Res/ResM-O/ResM*: Resnet-50 FPN (/w/ Oracle mask pooling/SAM mask pooling); *Eff/M-O/EffM*: EfficientNet-B0 FPN (/w/ Oracle mask pooling/SAM mask pooling).

## 3.3 DATASETS

We used the following datasets in this paper (see Table 2). Virtual KITTI is a synthetic clone of the KITTI tracking benchmark; unlike Cityscapes and KITTI Semantic Train, it comprises rendered video sequences rather than real-world still images.

## 3.4 CAUSAL ANALYSIS

In order to infer the FG-BG association by causal intervention, we design three separate experiments to empirically validate our hypothesis:

**Random BG assignment with FG:** While there could be associations between common objects such as "car" and "road", we are interested in evaluating if the car is detected in the absence of road. Therefore, we design a test to sample random BGs for each image frame and then placing the FGs into the BG image using a blending operation. The idea is then to check how many of these FGs are

Table 2: Summary of datasets and abbreviations used in our experiments

| Dataset | Description | Abbreviation(s) |
|---|---|---|
| Cityscapes (Cordts et al., 2016) | Urban scenes with 2975 train, 500 val, 1525 test images; 8 FG, 11 BG classes; includes Foggy and Rainy variants. | CV, CFV, CRV |
| KITTI Semantic Segmentation (Abu Alhaija et al., 2018) | Karlsruhe driving scenes; 200 train images, pixel-level labeling; 8 FG, 11 BG classes. | KST |
| Virtual KITTI (Gaidon et al., 2016) | Synthetic KITTI-based data; 6 weather conditions (only weather subset: 2188 images, 375×1242); 3 FG, 10 BG classes. | VKC, VKF, VKM, VKO, VKR, VKS |
| BG-20K (Li et al., 2022a) | 20,000 high-res background images, randomly combined with FG from other datasets. | BG-20K |

**Dataset abbreviations**:

*CV/CFV/CRV*: Cityscapes Validation/Foggy/Rainy; *KST*: KITTI Semantic Train; *BG-20K*: Background 20K Dataset;

*VKC/VKF/VKM*: Virtual KITTI Clone/Foggy/Morning; *VKO/VKR/VKS*: Virtual KITTI Overcast/Rain/Sunset.

detected given these modified images. Large-scale inference of such random placement can give us a measure of the robustness of the models. We evaluate a total 20 different datasets (3 Cityscapes related, "KST", and 6 Virtual KITTI related and with and without BG-20K compound images). The synthetic datasets consist of randomly chosen BG images compounded with FG images using FG masks. All images have different BG images.

**Fixed random BG assignment with FG:** While the previous experiment provides a distributional measure of the robustness, it does not provide a measure of the performance of a model for different FGs. Therefore, in this experiment, we utilize a single randomly chosen BG image for the 10 datasets and provide results on them (5 repetitions with 5 sampled BG images). This helps us provide a deterministic BG information on which to understand the performance of the model for different FGs.

**BG activation map perturbation:** We modify activation maps during inference and multiply a small weight to simulate BG perturbation in feature space. It demonstrates the robustness of trained models and the context bias between FG and BG. These studies (Ramaswamy et al., 2020; Meng et al., 2021) suggested FG activation is related to performance. We adjusted activation values using BG masks from 0.5 to 2.75 with a 0.25 step size.

### 3.5 MASK GENERATION PROCEDURE

Obtaining perfectly accurate masks is not practical. To address this, we use the SAM model to automatically generate incomplete masks. We observe an approximate error rate of 10–15% in pixel accuracy compared to ground truth masks. Such error levels are typical of small and simple off-the-shelf models, such as FCN-8s (Long et al., 2015), according to semantic segmentation benchmarks on datasets including Cityscapes, Pascal VOC 2012 (Everingham et al., 2015), and ADE20K (Zhou et al., 2017). In our experiments, "Oracle" models (denoted as "M-O") use ground truth masks, whereas "non-Oracle" models ("M") rely on incomplete masks generated by SAM. These experiments illustrate the upper bound achievable through FG-BG separation using a pooling layer. The number of masks used per dataset for our experiments is as follows: Cityscapes train (2975/2975), CV (500/500), CFV (1500/1500), CRV (1188/1188), KST (200/200), and Virtual KITTI (2188/2188).

## 4 EXPERIMENT

In this section, we provide the implementation details, including the evaluation metrics, number of evaluations and training parameters for each model. We utilize two different evaluation metrics:

- *mAP50* - Mean Average Precision at a threshold 50 is standard for detection tasks (Kay et al., 2025). We logged the mAP50 of each model for the different experiments to understand the trends.

- *Hierarchical F1 score* (Riehl et al., 2023) - In order to understand the effect of changing the BG activations or inputs, we need to provide a measure of change due to the intervention (how

many are detected after the intervention as opposed to before the intervention). Therefore, we select the hierarchical F1 score which provides a classification score for hierarchical problems.

We explain the training parameters in detail in Section A.1.

## 4.1 EXPERIMENTAL PROCEDURE

1. Evaluate 5 models including ALDI++ given by the official repository on 20 different validation set. Only "VKC" is a training set and validation set. To avoid the randomness effect, we measured 5 times of evaluation on the synthetic dataset for various BG.

2. Compute hierarchal F1 score variations on Cityscapees trained set and "VKC".

3. Measure changes of mAP50 depending on variations of BG activation perturbation.

## 4.2 AP50 MEASUREMENT EXPERIMENT RESULT

We remeasure the performance of ALDI++ mAP50 trained on Cityscapes training set, for "CRV" and "KST". ALDI++ outperforms other models on"CFV" and "CRV". It has the benefit of using target domain information obtained from Cityscapes foggy training set to learn foggy features.

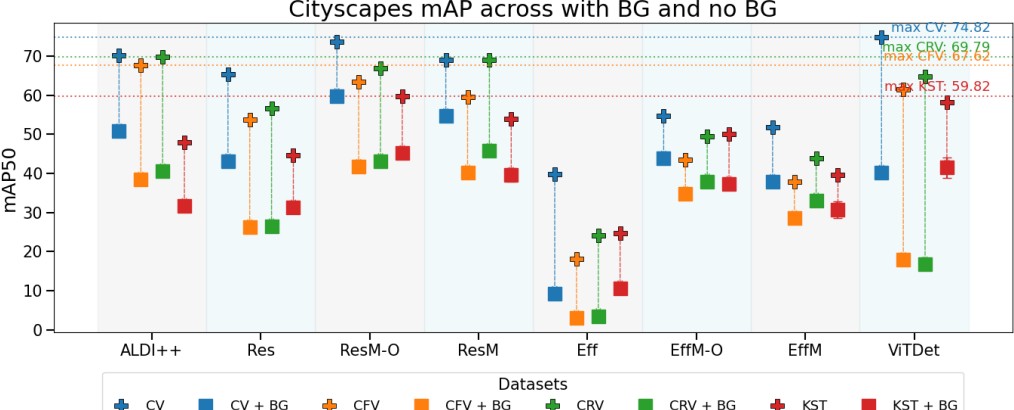

Figure 2: **mAP50 of 8 different models across 8 datasets.** The crosses indicate the results without background during evaluation and square shapes mean inference results with random background. The maximum mAP50 of each dataset is annotated in dashed lines. Standard deviation is displayed as error bars but it is trivial.

While "ResM-O" is superior to other models on "KST", "ViTDet" outperforms other models on "CV". Notably, "ResM-O" and "ResM" demonstrate superior outcomes on "KST" compared to ALDI++. "ResM" performs competitively on "CRV" compared to ALDI++. Even "EffM-O" beats ALDI++ by about 2% with the lowest computational costs among other models (see Figure 2). Significant improvement is captured between "Eff" and "EffM" with the lowest computational costs. 1024x2048 image resolution is used to measure GFLOPs and the number of parameters ("Params"). "Res" has 398.5 GFLOPs and 41.3M "Params", ALDI++, "ResM-O", and "ResM" have the same GFLOPs and "Params" with "Res". "Eff" has 237.7 GFLOPs and 20.6M "Params" and "EffM-O" and "EffM" show 236.7 GFLOPs and 20.6M "Params".

For the Cityscapes + "BG" datasets, "ResM-O" and "ResM" are superior to the other models on all datasets, including "ViTDet". "EffM-O" demonstrates a significant difference from ALDI++ on "KST+BG" about 5.5%. (see Figure 2). Moreover, "EffM-O" and "EffM" have been initialized on the Imagenet-1K pretrained model (Deng et al., 2009), which is less sustainable than COCO dataset (Lin et al., 2014) pretrained model. "ViTDet" demonstrates a significant performance drop compared to other models. Numerical numbers are denoted in Table 5, 6, and 7.

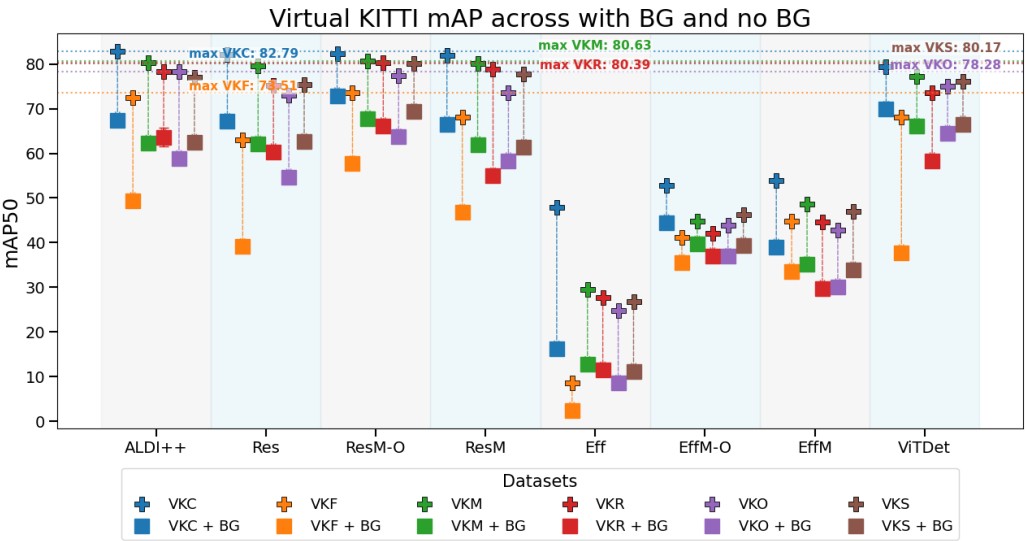

Figure 3: **mAP50 of 8 different models across 12 datasets.** "+" indicates without background during evaluation and square shapes mean inference results with random background. The maximum mAP50 of each dataset is annotated in dashed lines. Standard deviation is displayed as error bars but it is trivial.

With Virtual KITTI datasets, even though ALDI++ has the benefit of using target domain information ("VKF"), it cannot beat "ResM-O" on "VKF" (see Figure 3). "ResM-O" demonstrates its robustness across all synthetic datasets (see Figure 3). "ResM" shows competitive results to ALDI++ as well as "ViTDet". "EffM-O" cannot outperform ALDI++ in this case. "EffM" achieves significant performance gain compared to "Eff". mAP50 values are listed in 8, 9, and 10

Figure 4 displays mAP50 with all foreground objects on all images placed on single randomly chosen images from "BG-20K" (see Table 12 and 13 for precise mAP50). "ResM-O" is superior to other models. "ResM" achieves competitive results compared to "ALDI++" and surpasses "ViTDet" on Cityscapes datasets. It performs better than "ViTDet" on "VKF" which has a large domain shift from "VKC". "ViTDet" indicates it has more variations and significant accuracy drops on the random and fixed background experiments, which is different from our expectation. We believe that it is because "VitDet" architecture encodes the global context and thus, has larger FG-BG association. We perform a simple test of blocking certain BG classes such as $road$, $sidewalk$ etc., and measuring the performance drop (Fig. 6 in Section A.9). We note that the model has a significant drop for $road$, $sidewalk$ and $building$ which constitutes most of the training data BG distribution.

### 4.3 HIERARCHICAL F1 SCORE RESULTS

The computed average hierarchical F1 score, evaluated with BG modification, reflects individual performance gains from breaking the FG–BG association by comparing models trained with mask pooling versus without. Here, "DiffM-O" and "DiffM" denote the F1 score differences with the baseline model (e.g., "ResM") between "Oracle" model ("M-O") and mask-pooling models ("M"). The "M-O" models show substantial improvements in the number of foreground labels: "ResM-O" achieves 73/88 on Cityscapes train and 22/30 on "VKC", while "EffM-O" reaches 75/77 on Cityscapes and 26/30 on "VKC". Interestingly, the "M" models generally do not show significant improvements in F1 score, similar to mAP50, except for "EffM" on Cityscapes: "ResM" scores 47/88 on Cityscapes train and 14/30 on "VKC", while "EffM" achieves 62/77 on Cityscapes and 6/30 on "VKC". Notably, the mAP50 gains primarily arise from improvements on dominant objects such as $car$ and $person$. Tables 3 and 4 report the F1 score differences for each FG object.

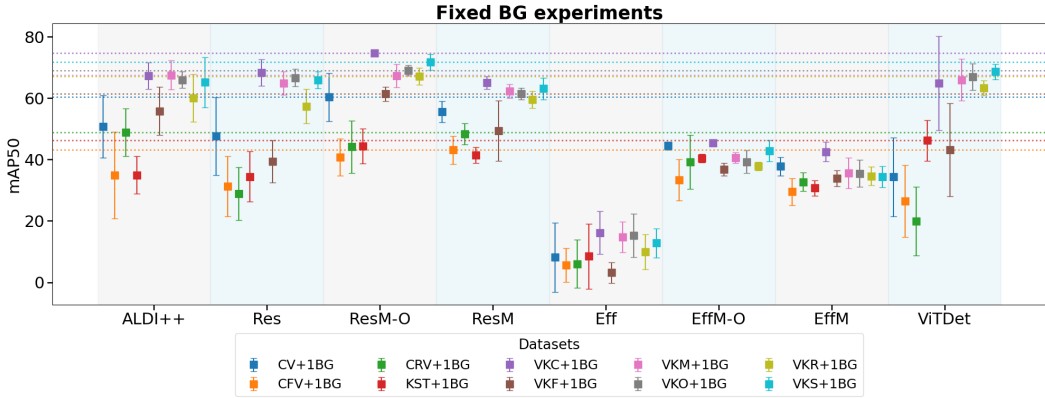

Figure 4: **mAP50 of 8 different models across 10 datasets.** Square shapes mean inference results with a fixed random background. The maximum mAP50 of each dataset is annotated in dashed lines. Standard deviation is displayed as an error bar at each marker.

### 4.4 BG ACTIVATION MAP PERTURBATION RESULTS

Overall, models with explicit FG-BG separation demonstrate their robustness on multiple domain shifted datasets (see Table 11). ALDI++ shows robustness on "CFV" and "CRV" because it has been trained with target domain information (Cityscapes Foggy train). However, "ResM-O" demonstrates comparable difference with respect to "CFV". It outperforms ALDI++ on "CV" and "KST" which ALDI++ has never seen. On Virtual KITTI, ALDI++ is superior to all models but "ResM-O" is dominant on Virtual KITTI + "BG" dataset, except on "VKC + BG" with the minuscule difference. "ResM" also demonstrates competitive results on the datasets mixed with random BG. It indicates FG-BG association separation during training and inference is helpful to enhance the capability of DAOD.

### 4.5 STATISTICAL ANALYSIS

We use two statistical tests: a paired bootstrap for the random-BG and fixed-BG experiments (Koehn, 2004), and the Wilcoxon signed-rank test for the BG-perturbation analysis (Conover, 1999). We evaluate the pairs ("Res", "ResM") and ("Eff", "EffM"), excluding the "M-O" variants because they are not practical. On Cityscapes and "KST," both pairs show statistically reliable gains under both random and fixed BG (see Table 18, 14). On Virtual KITTI with random BG, ("Res", "ResM") clears the threshold on 4/6 subsets, including "VKF," the largest domain shift, (see Table 16) while ("Eff", "EffM") does so on all 6/6 (see Table 17). With fixed BG, ("Res", "ResM") clears 3/6 (again including "VKF") (see Table 20), and ("Eff", "EffM") meets the criterion across Virtual KITTI under the fixed-BG setting (see Table 21). BG perturbation. Holding the foreground weight at 1.0, we vary the background activation from 0.5 to 2.75 in 0.25 steps and analyze AP50. On Cityscapes (with/without random-BG) and on "KST" (with/without random-BG) with BG perturbation, both pairs reject the null (see Table 11). On Virtual KITTI without random BG, ("Eff", "EffM") and ("Res", "ResM") each reach 5/6 subsets (see Table XX) and with random BG, ("Eff", "EffM") reaches 6/6 (see Table 11), whereas ("Res", "ResM") reaches 2/6 (see Table 11).

### 4.6 BENCHMARK FOR DAOD

Our mAP50 tables serve as a comprehensive benchmark for evaluating DAOD performance. ALDI++ is shown to achieve state-of-the-art mAP50 on the "CFV" dataset in their paper; however, it is not evaluated on "CRV" or "KST". Moreover, ALDI++ does not report results on the Virtual KITTI or the synthetic datasets constructed with random BGs. Notably, Virtual KITTI differs from standard object detection datasets, as it is originally designed for object tracking and exhibits strong temporal and spatial correlations across the dataset. This characteristic introduces a distinct evaluation perspective for DAOD.

## 5   DISCUSSION AND CONCLUSIONS

This work introduces a causally grounded analysis of context bias in DAOD with the pooling operation to alleviate under-examined spurious FG-BG associations. We analyze it with Mask Pooling as an intervention to explicitly separate FG and BG regions during feature aggregation where the mask could be provided as a ground truth and also provided by SAM, leading to improved generalization under domain shifts.

Our benchmark evaluations demonstrate that Mask Pooling with FG-BG separation consistently improves robustness under domain shifts, especially under completely random BGs (e.g., BG-20K). In particular, "M-O" models demonstrate substantial improvements in hierarchical F1 scores over its baseline, achieving gains in evaluation pairs (see Section 4.3). These improvements of EfficientNet are more pronounced than those observed between "Res" and "ResM-O", likely due to EfficientNet's architectural characteristics: it lacks early-stage max pooling and uses depthwise convolutions, resulting in larger effective receptive fields. This may amplify the impact of mask pooling with FG-BG separation, particularly for suppressing background features. However, it is notable that "EffM" demonstrates significant improvement with the lowest computational costs.

Our evaluation on the Virtual KITTI dataset reveals the effects of spatial and temporal correlations, as the BG and FG structures remain consistent across adjacent frames and scenes. ALDI++, having been trained with access to target domain information including BG features, leverages these correlations effectively. However, "ResM-O" and "ResM", trained without such access, demonstrates superiority even when BG perturbations (via BG-20K) disrupt FG-BG association. This highlights that models relying on spurious context may perform well under correlated conditions but generalize poorly in randomized settings. Although mask pooling with FG-BG separation improves F1 scores in most cases, the gains are not uniform across all classes. For example, while "M-O" models exhibited significant improvements across nearly all FG objects, "M" models returned marginal or even negative differences. This variation suggests that certain object classes may inherently depend more on contextual cues, or that architectural differences impact pooling effectiveness across object categories to make synergy.

Our findings indicate that ALDI++ learns strong FG-BG associations, likely due to its multi-stage training on foggy variants of Cityscapes. Its insufficient robustness under BG perturbation and high performance on Virtual KITTI support this, reinforcing the idea that models trained with target domain context implicitly encode BG priors. The performance of "ViTDet" is intriguing. In contrast to its global feature aggregation benefit, it returns significant accuracy drops on the BG image robustness experiments. Specific classes such as $road$ and $sidewalk$ have major accuracy drops (See Appendix A.9).

Context bias can be derived from imbalance of classes in datasets (LaBonte et al., 2024) suggests mixture balancing to alleviate the imbalance. For example, the Cityscapes dataset is biased to $road$ pixels. However, it is questionable whether it is achievable to obtain rare classes and compute a balance before training.

The FG-BG association is not universally spurious. Background context can aid recognition—e.g., vision–language models that leverage scene cues improve classification (Janouskova et al., 2024). We agree that background information can be beneficial, but in domain adaptation, it should be incorporated properly, separating causal context from domain-specific correlations so that gains transfer across domains (Son & Kusari, 2024).

### 5.1   LIMITATIONS

In this analytical work, we utilize the ground-truth masks or generate the FG mask using SAM, which is then used to split the FG-BG and perform object detection. A major limitation, and one we hope to work on as part of future work, is to have the boundary generation and/or correction process be a part of the training procedure. In such a scenario, the boundary is first estimated and then utilized in the detection process. It can cause latency and computational cost due to additional inference. We are also aware that the pooling method cannot be directly applied to "ViTDeT" because of its architecture. We aim to investigate this as well in our future work.

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

# A APPENDIX

## A.1 TRAINING PARAMETERS

We utilize the Detectron2 framework to train the models described above on the Cityscapes training set, other than ALDI++ which comes pretrained on Cityscapes. For Virtual KITTI datasets, we train our models with the same training parameters while ALDI++ is trained with "VKC" as source domain and "VKF" as target domain using the best training strategy. We use a learning rate of 0.02 for "Res", "ResM", and "ResM-O" and 0.01 for "Eff", "EffM", and "EffM-O" with a max iteration of 10000 for "Res", "ResM", and "ResM-O" and 20000 for "Eff", "EffM", and "EffM-O". For ALDI++, all training parameters are maintained except the maximum iteration. We apply the different data augmentation techniques for burn-in in ALDI++ except cutout (DeVries & Taylor, 2017). We used a batch size of 8 with an image resolution of 1024x2048. We train and evaluate the model on NVIDIA RTX A4500 20GB without any change in loss function. "ViTDet" is trained on NVIDIA RTX A4600 with a batch size of 2 for Cityscapes and 3 for Virtual KITTI. A learning rate of 0.0005 is used with 20000 iterations for Cityscapes and 15000 iterations for Virtual KITTI.

## A.2 HIERARCHICAL F1 RESULTS

Table 3: **Hierarchical F1 score improvements across different backbones on Cityscapes train set**. Dashes denote missing values.

| FG | Res | ResM-O | ResM | DiffM-0 | DiffM | Eff | EffM-O | EffM | DiffM-0 | DiffM |
|---|---|---|---|---|---|---|---|---|---|---|
| bicycle | 0.67 | 0.69 | 0.57 | 0.01 | -0.10 | 0.32 | 0.48 | 0.46 | 0.16 | 0.13 |
| bus | 0.68 | 0.74 | 0.63 | 0.05 | -0.06 | 0.07 | 0.37 | 0.34 | 0.30 | 0.27 |
| car | 0.81 | 0.86 | 0.84 | 0.05 | 0.02 | 0.75 | 0.79 | 0.80 | 0.03 | 0.05 |
| motorcycle | 0.62 | 0.59 | 0.61 | -0.03 | -0.01 | 0.31 | 0.44 | 0.31 | 0.13 | -0.001 |
| person | 0.67 | 0.80 | 0.69 | 0.13 | 0.02 | 0.40 | 0.66 | 0.63 | 0.26 | 0.22 |
| rider | 0.67 | 0.68 | 0.67 | 0.006 | -0.008 | 0.52 | 0.62 | 0.47 | 0.09 | -0.05 |
| train | 0.36 | 0.82 | 0.66 | 0.46 | 0.30 | - | - | - | - | - |
| truck | 0.58 | 0.64 | 0.64 | 0.05 | 0.05 | 0.02 | 0.24 | 0.22 | 0.22 | 0.20 |

Table 4: **Hierarchical F1 score improvements across different backbones on "VKC"**.

| FG | Res | ResM-O | ResM | DiffM-O | DiffM | Eff | EffM-O | EffM | DiffM-O | DiffM |
|---|---|---|---|---|---|---|---|---|---|---|
| Car | 0.75 | 0.75 | 0.75 | -0.001 | -0.006 | 0.58 | 0.59 | 0.55 | 0.009 | -0.03 |
| Truck | 0.96 | 0.95 | 0.96 | -0.01 | -0.000 | 0.69 | 0.70 | 0.67 | 0.01 | -0.02 |
| Van | 0.83 | 0.82 | 0.83 | -0.006 | -0.001 | 0.55 | 0.61 | 0.54 | 0.05 | -0.01 |

## A.3 MAP50 CITYSCAPES

Table 5: **mAP50 scores on various Cityscapes datasets.** "reported" means the value from the published paper.

| Dataset | ALDI++ | Res | ResM-O | ResM | Eff | EffM-O | EffM | ViTDet |
|---|---|---|---|---|---|---|---|---|
| CV | 70.09 (reported:NA) | 65.38 | 73.57 | 69.00 | 39.88 | 54.72 | 51.72 | **74.82** |
| CFV | **67.62** (reported: 66.8) | 53.77 | 63.41 | 59.57 | 18.17 | 43.49 | 37.96 | 61.46 |
| CRV | **69.79** (reported: NA) | 56.55 | 66.97 | 69.04 | 24.25 | 49.52 | 43.78 | 64.75 |
| KST | 47.96 (reported: NA) | 44.67 | **59.82** | 53.99 | 24.79 | 50.02 | 39.59 | 58.25 |

Table 6: **Mean ± standard deviation of mAP50 (Group 1: ALDI++ and ResNet models).**

| Dataset | ALDI++ | Res | ResM-O | ResM |
|---|---|---|---|---|
| CV + BG | 50.74 ± 1.02 | 43.10 ± 1.09 | **59.80** ± 1.10 | 54.65 ± 0.33 |
| CFV + BG | 38.42 ± 0.40 | 26.29 ± 0.21 | **41.67** ± 0.07 | 40.27 ± 0.52 |
| CRV + BG | 40.52 ± 0.27 | 26.49 ± 0.45 | 43.16 ± 0.20 | **45.75** ± 0.55 |
| KST + BG | 31.66 ± 1.09 | 31.29 ± 0.93 | **45.20** ± 0.38 | 39.63 ± 1.80 |

Table 7: **Mean ± standard deviation of mAP50 (Group 2: EfficientNet and ViT models).**

| Dataset | Eff | EffM-O | EffM | ViTDet |
|---|---|---|---|---|
| CV + BG | 9.21 ± 0.37 | 43.80 ± 0.35 | 37.85 ± 0.83 | 40.18 ± 1.32 |
| CFV + BG | 3.09 ± 0.19 | 34.73 ± 0.29 | 28.55 ± 0.25 | 17.89 ± 0.54 |
| CRV + BG | 3.53 ± 0.21 | 37.97 ± 0.43 | 33.09 ± 0.05 | 16.76 ± 0.90 |
| KST + BG | 10.66 ± 1.05 | 37.37 ± 1.30 | 30.80 ± 2.13 | 41.49 ± 2.58 |

## A.4   MAP50 VIRTUAL KITTI

Table 8: **mAP50 scores on various Virtual KITTI.**

| Dataset | ALDI++ | Res | ResM-O | ResM | Eff | EffM-O | EffM | ViTDet |
|---|---|---|---|---|---|---|---|---|
| VKC | **82.79** | 82.05 | 82.41 | 82.04 | 47.87 | 52.81 | 53.84 | 79.42 |
| VKF | 72.49 | 63.05 | **73.51** | 68.04 | 8.52 | 41.09 | 44.86 | 68.04 |
| VKM | 80.36 | 79.62 | **80.63** | 80.07 | 29.54 | 44.84 | 48.58 | 77.18 |
| VKR | 78.35 | 75.23 | **80.39** | 78.79 | 27.66 | 42.11 | 44.56 | 74.97 |
| VKO | **78.28** | 73.09 | 77.40 | 73.51 | 24.72 | 43.86 | 42.74 | 73.55 |
| VKS | 77.00 | 75.35 | **80.17** | 77.71 | 26.81 | 46.24 | 46.96 | 76.12 |

Table 9: **Mean ± standard deviation of mAP50 with random BG (Group 1: ALDI++ and ResNet family).**

| Dataset | ALDI++ | Res | ResM-O | ResM |
|---|---|---|---|---|
| VKC + BG | 67.45 ± 0.18 | 67.14 ± 0.16 | **72.90** ± 0.35 | 66.52 ± 0.47 |
| VKF + BG | 49.44 ± 0.21 | 39.11 ± 0.56 | **57.66** ± 0.28 | 46.78 ± 0.82 |
| VKM + BG | 62.28 ± 0.25 | 62.08 ± 0.34 | **67.77** ± 0.67 | 61.95 ± 0.60 |
| VKR + BG | 63.64 ± 2.10 | 60.36 ± 0.42 | **66.19** ± 0.40 | 54.95 ± 0.39 |
| VKO + BG | 58.83 ± 0.49 | 54.69 ± 0.71 | **63.69** ± 0.54 | 58.35 ± 0.79 |
| VKS + BG | 62.45 ± 0.21 | 62.62 ± 0.24 | **69.32** ± 0.23 | 61.34 ± 0.18 |

Table 10: **Mean ± standard deviation of mAP50 with random BG (Group 2: EfficientNet family and ViTDet).**

| Dataset | Eff | EffM-O | EffM | ViTDet |
|---|---|---|---|---|
| VKC + BG | 16.19 ± 0.18 | 44.46 ± 0.29 | 39.01 ± 0.36 | 69.85 ± 0.33 |
| VKF + BG | 2.45 ± 0.11 | 35.52 ± 0.11 | 33.55 ± 0.58 | 37.76 ± 0.89 |
| VKM + BG | 12.69 ± 0.47 | 39.64 ± 0.45 | 35.08 ± 0.27 | 66.03 ± 0.46 |
| VKR + BG | 11.48 ± 0.13 | 37.04 ± 0.30 | 29.76 ± 0.64 | 58.36 ± 0.46 |
| VKO + BG | 8.50 ± 0.11 | 36.92 ± 0.24 | 30.11 ± 0.44 | 64.47 ± 0.56 |
| VKS + BG | 11.16 ± 0.16 | 39.38 ± 0.31 | 33.89 ± 0.22 | 66.57 ± 0.29 |

Table 11: **mAP50 range comparison (Min / Max / Diff) across 20 datasets with and without BG intervention.** Bold values indicate the lowest variation (Diff) in each dataset group. Lower Diff indicates robustness on BG perturbation. Italic values mean lower Diff of "EffM-O" than ALDI++. † and ∗ indicate statistically significant difference for each pair and dataset.

| Model | CV | | | CFV | | | CRV | | | KST | | |
|---|---|---|---|---|---|---|---|---|---|---|---|---|
| | Min | Max | Diff | Min | Max | Diff | Min | Max | Diff | Min | Max | Diff |
| ALDI++ | 63.07 | 71.82 | 8.75 | 59.05 | 67.64 | **8.60** | 67.13 | 71.11 | **3.98** | 25.87 | 50.32 | 24.45 |
| EffM-O | 10.46 | 54.73 | 44.27 | 13.10 | 43.39 | 30.29 | 4.93 | 50.50 | 45.57 | 24.68 | 52.72 | 28.04 |
| ResM-O | 66.27 | 73.57 | **7.30** | 54.75 | 63.60 | 8.85 | 60.12 | 66.97 | 6.85 | 43.32 | 60.19 | 16.86 |
| Eff | 0.90 | 21.30 | 20.40* | 1.78 | 26.88 | 25.10* | 2.46 | 44.05 | 41.59* | 0.40 | 30.82 | 30.42* |
| EffM | 35.69 | 51.74 | 16.06* | 26.19 | 38.80 | 12.60* | 34.06 | 43.60 | 9.54* | 21.47 | 41.13 | 19.66* |
| Res | 54.75 | 65.49 | 10.74† | 38.19 | 54.53 | 16.34† | 39.76 | 56.70 | 16.94† | 21.54 | 44.70 | 23.16† |
| ResM | 61.57 | 69.59 | 8.02† | 50.40 | 59.34 | 8.93† | 59.62 | 68.97 | 9.35† | 36.82 | 53.02 | **16.20†** |
| Model | CV+BG | | | CFV+BG | | | CRV+BG | | | KST+BG | | |
| | Min | Max | Diff | Min | Max | Diff | Min | Max | Diff | Min | Max | Diff |
| ALDI++ | 30.74 | 58.75 | 28.01 | 22.95 | 45.29 | 22.34 | 27.87 | 46.44 | 18.58 | 15.66 | 38.30 | 22.63 |
| EffM-O | 22.16 | 44.45 | *22.28* | 18.19 | 34.44 | *16.25* | 22.49 | 39.83 | *17.34* | 20.58 | 41.03 | *20.45* |
| ResM-O | 42.06 | 62.23 | 20.17 | 26.60 | 45.19 | 18.59 | 26.81 | 46.54 | 19.73 | 28.23 | 45.51 | **17.28** |
| Eff | 0.14 | 9.27 | 9.13* | 0.13 | 10.64 | 10.50* | 0.38 | 19.08 | 18.70* | 1.60 | 17.71 | 16.10* |
| EffM | 21.75 | 40.08 | 18.34* | 16.17 | 28.74 | 12.56* | 18.35 | 33.74 | 15.39* | 15.63 | 29.35 | 13.72* |
| Res | 23.78 | 52.55 | 28.77† | 15.62 | 34.22 | 18.60† | 14.89 | 33.50 | 18.60† | 12.75 | 35.70 | 22.95† |
| ResM | 41.71 | 58.31 | **16.60†** | 29.69 | 42.62 | **12.93†** | 34.05 | 49.37 | **15.32†** | 22.82 | 41.17 | 18.35† |
| Model | VKC | | | VKF | | | VKM | | | VKO | | |
| | Min | Max | Diff | Min | Max | Diff | Min | Max | Diff | Min | Max | Diff |
| ALDI++ | 76.69 | 82.30 | **5.61** | 57.30 | 72.49 | **15.19** | 71.53 | 80.37 | **8.84** | 71.87 | 78.36 | **6.49** |
| EffM-O | 33.05 | 52.81 | 19.76 | 23.28 | 41.10 | 17.82 | 30.35 | 45.43 | 15.08 | 29.77 | 43.05 | 13.28 |
| ResM-O | 72.81 | 82.41 | 9.61 | 43.87 | 73.51 | 29.64 | 69.87 | 80.63 | 10.76 | 66.62 | 80.39 | 13.78 |
| Eff | 23.79 | 48.03 | 24.24* | 3.03 | 8.54 | 5.52 | 8.59 | 31.68 | 23.09* | 18.45 | 28.52 | 10.06* |
| EffM | 7.53 | 53.96 | 46.43* | 12.11 | 44.84 | 32.73 | 11.22 | 48.90 | 37.68* | 22.67 | 44.12 | 21.45* |
| Res | 67.63 | 82.07 | 14.44† | 18.55 | 63.13 | 44.57† | 56.61 | 79.63 | 23.02† | 56.34 | 75.68 | 19.34† |
| ResM | 72.90 | 82.05 | 9.15† | 41.43 | 67.64 | 26.21† | 65.75 | 80.10 | 14.35† | 62.71 | 78.85 | 16.14† |
| Model | VKR | | | VKS | | | VKC+BG | | | VKF+BG | | |
| | Min | Max | Diff | Min | Max | Diff | Min | Max | Diff | Min | Max | Diff |
| ALDI++ | 64.63 | 78.29 | **13.65** | 66.90 | 77.00 | **10.10** | 49.25 | 70.61 | 21.36 | 25.40 | 55.86 | 30.46 |
| EffM-O | 32.17 | 43.87 | 11.70 | 29.93 | 46.24 | 16.31 | 17.28 | 44.47 | 27.19 | 11.26 | 35.50 | *24.24* |
| ResM-O | 54.54 | 77.41 | 22.87 | 69.23 | 80.18 | 10.95 | 51.13 | 72.83 | 21.70 | 31.40 | 58.72 | 27.32 |
| Eff | 10.44 | 24.74 | 14.30* | 12.82 | 27.47 | 14.65* | 3.07 | 20.43 | 17.36* | 0.41 | 4.03 | 3.63* |
| EffM | 25.98 | 44.42 | 18.44* | 10.18 | 48.33 | 38.16* | 12.66 | 40.55 | 27.90* | 9.34 | 34.69 | 25.35* |
| Res | 35.88 | 73.91 | 38.03 | 54.17 | 75.54 | 21.37† | 35.01 | 69.13 | 34.12 | 13.84 | 47.41 | 33.57† |
| ResM | 50.95 | 73.39 | 22.44 | 64.60 | 77.33 | 12.73† | 48.36 | 67.41 | **19.05** | 26.28 | 49.23 | **22.94†** |
| Model | VKM+BG | | | VKO+BG | | | VKR+BG | | | VKS+BG | | |
| | Min | Max | Diff | Min | Max | Diff | Min | Max | Diff | Min | Max | Diff |
| ALDI++ | 39.75 | 66.86 | 27.11 | 41.54 | 66.09 | 24.55 | 35.93 | 63.80 | 27.87 | 39.86 | 66.84 | 26.98 |
| EffM-O | 16.48 | 39.62 | *23.14* | 14.12 | 37.03 | *22.91* | 13.17 | 36.90 | *23.73* | 15.05 | 40.02 | *24.97* |
| ResM-O | 45.14 | 68.11 | 22.97 | 40.48 | 65.88 | 25.40 | 37.84 | 64.12 | 26.28 | 45.90 | 69.29 | **23.39** |
| Eff | 1.97 | 16.68 | 14.71* | 1.89 | 14.36 | 12.47* | 1.28 | 11.21 | 9.93* | 1.75 | 14.53 | 12.78* |
| EffM | 11.48 | 35.90 | 24.42* | 10.87 | 32.14 | 21.27* | 9.52 | 31.34 | 21.82* | 10.63 | 34.99 | 24.35* |
| Res | 30.38 | 64.70 | 34.32 | 26.38 | 62.66 | 36.28 | 21.29 | 57.52 | 36.23† | 28.51 | 64.61 | 36.09 |
| ResM | 40.78 | 62.81 | **22.04** | 36.35 | 60.58 | **24.23** | 34.15 | 56.91 | **22.76†** | 39.76 | 63.23 | 23.47 |

## A.5 MAP50 COMPARISON BASED ON BG INTERVENTION

Table 12: **Mean ± standard deviation of mAP50 with random BG (Group 1: ALDI++ and ResNet family).** Bold values mean the highest mAP50 across other models.

| Dataset | ALDI++ | Res | ResM-O | ResM |
|---|---|---|---|---|
| CV + 1 BG | $50.79 \pm 10.13$ | $47.63 \pm 12.66$ | $\mathbf{60.31} \pm 7.82$ | $55.55 \pm 3.49$ |
| CFV + 1 BG | $34.91 \pm 14.09$ | $31.26 \pm 9.86$ | $\mathbf{40.76} \pm 6.06$ | $43.08 \pm 4.57$ |
| CRV + 1 BG | $\mathbf{48.86} \pm 7.78$ | $28.91 \pm 8.60$ | $44.11 \pm 8.59$ | $48.39 \pm 3.42$ |
| KST + 1 BG | $34.96 \pm 6.09$ | $34.43 \pm 8.18$ | $\mathbf{44.36} \pm 5.71$ | $41.41 \pm 2.57$ |
| VKC + 1 BG | $67.24 \pm 4.32$ | $68.27 \pm 4.34$ | $\mathbf{74.72} \pm 0.99$ | $65.06 \pm 2.04$ |
| VKF + 1 BG | $55.76 \pm 7.85$ | $39.39 \pm 6.90$ | $\mathbf{61.35} \pm 2.34$ | $49.29 \pm 9.81$ |
| VKM + 1 BG | $\mathbf{67.51} \pm 4.76$ | $64.86 \pm 3.86$ | $67.32 \pm 3.77$ | $62.27 \pm 2.24$ |
| VKO + 1 BG | $65.98 \pm 2.61$ | $66.64 \pm 2.81$ | $\mathbf{69.02} \pm 1.66$ | $61.39 \pm 1.86$ |
| VKR + 1 BG | $60.00 \pm 7.75$ | $57.33 \pm 5.61$ | $\mathbf{67.14} \pm 2.74$ | $59.53 \pm 2.72$ |
| VKS + 1 BG | $65.19 \pm 8.17$ | $65.91 \pm 2.76$ | $\mathbf{71.76} \pm 2.54$ | $63.10 \pm 3.53$ |

Table 13: **Mean ± standard deviation of mAP50 with random BG (Group 2: EfficientNet family and ViTDet).**

| Dataset | Eff | EffM-O | EffM | ViTDet |
|---|---|---|---|---|
| CV + 1 BG | $8.16 \pm 11.30$ | $44.48 \pm 1.29$ | $37.76 \pm 3.00$ | $34.35 \pm 12.85$ |
| CFV + 1 BG | $5.59 \pm 5.50$ | $33.33 \pm 6.77$ | $29.48 \pm 4.40$ | $26.47 \pm 11.70$ |
| CRV + 1 BG | $6.02 \pm 7.78$ | $39.17 \pm 8.75$ | $32.70 \pm 3.02$ | $19.96 \pm 11.18$ |
| KST + 1 BG | $8.51 \pm 10.61$ | $\mathit{40.43} \pm 1.38$ | $30.69 \pm 2.48$ | $46.24 \pm 6.63$ |
| VKC + 1 BG | $16.19 \pm 6.99$ | $45.44 \pm 0.86$ | $42.53 \pm 3.23$ | $64.80 \pm 15.32$ |
| VKF + 1 BG | $3.13 \pm 3.37$ | $36.75 \pm 2.11$ | $33.91 \pm 2.56$ | $43.21 \pm 15.15$ |
| VKM + 1 BG | $14.68 \pm 4.98$ | $40.55 \pm 1.73$ | $35.56 \pm 5.02$ | $65.96 \pm 6.84$ |
| VKO + 1 BG | $15.21 \pm 7.07$ | $39.28 \pm 3.72$ | $35.48 \pm 4.42$ | $66.89 \pm 4.34$ |
| VKR + 1 BG | $9.87 \pm 5.71$ | $37.86 \pm 1.42$ | $34.63 \pm 3.08$ | $63.37 \pm 2.41$ |
| VKS + 1 BG | $12.78 \pm 4.74$ | $42.86 \pm 3.46$ | $34.40 \pm 3.49$ | $68.59 \pm 2.49$ |

## A.6 STATISTICAL ANALYSIS FOR RANDOM BG EXPERIMENT

It describes statistical analysis between baseline models ("Res" and "Eff") and SAM models ("ResM" and "EffM")

Table 14: **Results of statistical analysis on Cityscapes + random BG.** "Res" vs "ResM" (paired bootstrap). Diff = "ResM" - "Res". Confidence Intervals (CIs) are 95%.

| Dataset | "Res" mean | "ResM" mean | Diff | CI low | CI high | p_value | Rel. gain |
|---|---|---|---|---|---|---|---|
| CV | 42.58 | 54.65 | 12.06 | 11.56 | 12.63 | $\approx 0$ | 28.3% |
| CFV | 26.34 | 40.27 | 13.93 | 13.46 | 14.44 | $\approx 0$ | 52.9% |
| CRV | 26.212 | 45.75 | 19.54 | 19.34 | 19.74 | $\approx 0$ | 74.6% |
| KST | 31.62 | 39.63 | 8.00 | 6.87 | 8.94 | $\approx 0$ | 25.3% |

Table 15: **Results of statistical analysis on Cityscapes + random BG.** "Eff" vs "EffM" (paired bootstrap). Diff = "EffM" − "Eff". CIs are 95%.

| Dataset | "Eff" mean | "EffM" mean | Diff | CI low | CI high | p_value | Rel. gain |
|---------|-----------|------------|-------|--------|---------|---------|-----------|
| CV | 9.21 | 37.85 | 28.64 | 28.09 | 29.20 | ≈ 0 | 311.0% |
| CFV | 3.09 | 28.55 | 25.46 | 25.14 | 25.76 | ≈ 0 | 823.2% |
| CRV | 3.53 | 33.09 | 29.56 | 29.35 | 29.76 | ≈ 0 | 837.1% |
| KST | 10.66 | 30.81 | 20.14 | 18.80 | 21.410 | ≈ 0 | 189.0% |

Table 16: **Results of statistical analysis on Virtual KITTI + random BG.** "Res" vs "ResM" (paired bootstrap). Diff = "ResM" − "Res". CIs are 95%.

| Dataset | "Res" mean | "ResM" mean | Diff | CI low | CI high | p_value | Rel. gain |
|---------|-----------|------------|-------|--------|---------|---------|-----------|
| VKC | 67.14 | 66.92 | -0.21 | -0.68 | 0.24 | 0.36 | -0.3% |
| VKF | 39.10 | 46.78 | 7.67 | 7.08 | 8.26 | ≈ 0 | 19.6% |
| VKM | 61.87 | 61.95 | 0.07 | -0.51 | 0.69 | 0.78 | 0.1% |
| VKO | 60.36 | 58.79 | -1.56 | -2.09 | -1.04 | ≈ 0 | -2.6% |
| VKR | 54.69 | 55.35 | 0.66 | 0.13 | 1.17 | 0.016 | 1.2% |
| VKS | 62.62 | 61.34 | -1.28 | -1.89 | -0.69 | ≈ 0 | -2.0% |

Table 17: **Results of statistical analysis on Virtual KITTI + random BG.** "Eff" vs "EffM" (paired bootstrap). Diff = "EffM" − "Eff". CIs are 95%.

| Dataset | "Eff" mean | "EffM" mean | Diff | CI low | CI high | p_value | Rel. gain |
|---------|-----------|------------|-------|--------|---------|---------|-----------|
| VKC | 16.19 | 39.01 | 22.81 | 22.52 | 23.11 | ≈ 0 | 141.0% |
| VKF | 2.45 | 33.15 | 30.70 | 30.51 | 30.89 | ≈ 0 | 1252.2% |
| VKM | 12.69 | 35.08 | 22.39 | 22.11 | 22.66 | ≈ 0 | 176.4% |
| VKO | 11.47 | 30.51 | 19.03 | 18.79 | 19.28 | ≈ 0 | 165.9% |
| VKR | 8.50 | 29.36 | 20.86 | 20.68 | 21.03 | ≈ 0 | 245.6% |
| VKS | 11.16 | 33.79 | 22.63 | 22.43 | 22.83 | ≈ 0 | 202.9% |

## A.7 STATISTICAL ANALYSIS FOR FIXED BG EXPERIMENT

It describes statistical analysis between baseline models ("Res" and "Eff") and SAM models ("ResM" and "EffM")

Table 18: **Results of statistical analysis on Cityscapes + fixed BG.** "Res" vs "ResM" (paired bootstrap). Diff = "ResM" − "Res". CIs are 95%.

| Dataset | "Res" mean | "ResM" mean | Diff | CI low | CI high | p_value | Rel. gain |
|---------|-----------|------------|-------|--------|---------|---------|-----------|
| CV | 45.45 | 55.95 | 10.50 | 6.28 | 14.65 | ≈ 0 | 23.1% |
| CFV | 28.81 | 43.08 | 14.27 | 9.600 | 18.86 | ≈ 0 | 49.5% |
| CRV | 26.77 | 48.41 | 21.64 | 16.779 | 26.22 | ≈ 0 | 80.8% |
| KST | 31.41 | 41.41 | 9.99 | 2.190 | 17.59 | 0.012 | 31.8% |

Table 19: **Results of statistical analysis on Cityscapes + fixed BG.** "Eff" vs "EffM" (paired bootstrap). Diff = "EffM" − "Eff". CIs are 95%.

| Dataset | "Eff" mean | "EffM" mean | Diff | CI low | CI high | p_value | Rel. gain |
|---------|-----------|------------|-------|--------|---------|---------|-----------|
| CV | 9.11 | 37.76 | 28.64 | 19.60 | 34.72 | ≈ 0 | 314.4% |
| CFV | 5.28 | 29.48 | 24.20 | 22.30 | 25.55 | ≈ 0 | 458.0% |
| CRV | 6.02 | 32.70 | 26.68 | 23.25 | 29.36 | ≈ 0 | 442.9% |
| KST | 16.75 | 30.69 | 13.94 | 7.98 | 21.01 | ≈ 0 | 83.2% |

Table 20: **Results of statistical analysis on Virtual KITTI + fixed BG.** "Res" vs "ResM" (paired bootstrap). Diff = "ResM" − "Res". CIs are 95%.

| Dataset | "Res"mean | "ResM" mean | Diff | CI low | CI high | p_value | Rel. gain |
|---------|-----------|-------------|------|--------|---------|---------|-----------|
| VKC | 66.94 | 65.06 | -1.87 | -5.56 | 1.37 | 0.29 | -2.8% |
| VKF | 38.98 | 49.29 | 10.31 | 0.29 | 20.78 | 0.04 | 26.5% |
| VKM | 63.50 | 62.27 | -1.22 | -4.16 | 1.61 | 0.43 | -1.9% |
| VKO | 66.06 | 61.39 | -4.67 | -7.41 | -1.70 | 0.00 | -7.1% |
| VKR | 55.22 | 59.53 | 4.30 | 0.61 | 7.97 | 0.02 | 7.8% |
| VKS | 64.77 | 63.10 | -1.66 | -4.57 | 1.38 | 0.28 | -2.6% |

Table 21: **Results of statistical analysis on Virtual KITTI + fixed BG.** "Eff" vs "EffM" (paired bootstrap). Diff = "EffM" "Eff". CIs are 95%.

| Dataset | "Eff" mean | "EffM" mean | Diff | CI low | CI high | p_value | Rel. gain |
|---------|------------|-------------|------|--------|---------|---------|-----------|
| VKC | 18.51 | 42.53 | 24.01 | 20.93 | 28.67 | $\approx 0$ | 129.7% |
| VKF | 4.83 | 33.91 | 29.08 | 25.17 | 33.33 | $\approx 0$ | 601.5% |
| VKM | 14.63 | 35.56 | 20.93 | 18.04 | 23.83 | $\approx 0$ | 143.1% |
| VKO | 16.16 | 35.48 | 19.32 | 12.67 | 25.97 | $\approx 0$ | 119.6% |
| VKR | 12.26 | 34.63 | 22.36 | 20.03 | 25.27 | $\approx 0$ | 182.4% |
| VKS | 10.92 | 34.40 | 23.48 | 20.54 | 27.24 | $\approx 0$ | 215.1% |

### A.8 STATISTICAL ANALYSIS FOR BG PERTURBATION EXPERIMENT

It describes statistical analysis between baseline models ("Res" and "Eff") and SAM models ("ResM" and "EffM") (see Table 11).

### A.9 IMAGE SPACE DROP RATE FOR "VITDET"

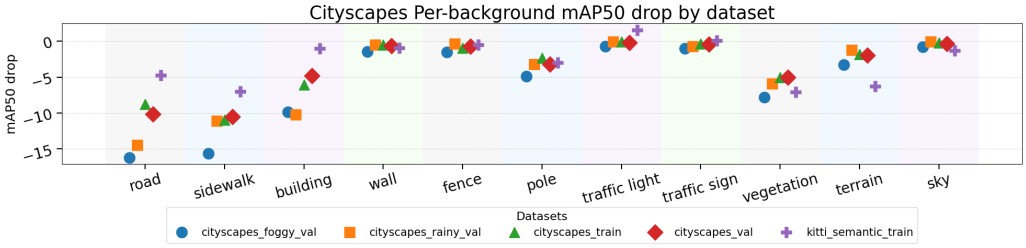

Figure 5: The figure depicts the mAP50 drop of "ViTDet" on the Cityscapes datasets and "KST". *Road* has a significant association with the mAP50. Without *trafficlight*, mAP50 of "KST" increases.

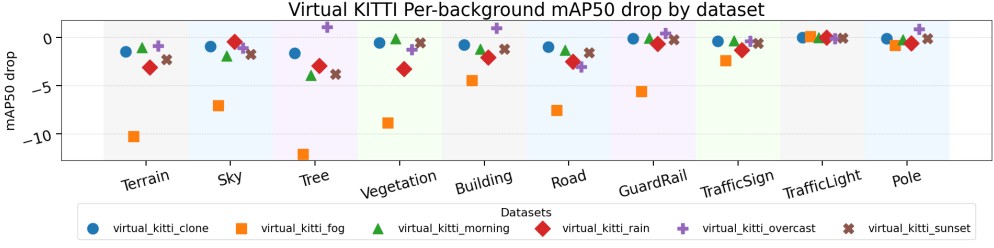

Figure 6: The figure illustrates the mAP50 drop of "ViTDet" on the Virtual KITTI datasets. $Tree$ has a significant association with the mAP50. Absence of several BG labels increases mAP50 for "VKO".

