# OpenReview forum: "Mitigating Context Bias via Foreground-Background Separation and Pooling: A Causal Analysis and Robust Evaluation"
_ICLR.cc/2026/Conference — ICLR 2026 Conference Withdrawn Submission_

### Official Review · Reviewer_tfiX · 2025-10-31

**Soundness:** 2
**Presentation:** 2
**Contribution:** 2
**Rating:** 4
**Confidence:** 3

**Summary:**

This paper investigates the issue of context bias in domain adaptation for object detection (DAOD), where models inadvertently learn spurious associations between foreground objects and their backgrounds during training. The authors propose a causal perspective to understand and mitigate this bias and introduce an analytical framework based on explicit foreground-background separation using a Mask Pooling operation, which leverages either ground truth or SAM-generated masks to isolate object features during feature aggregation.

**Strengths:**

1. The approach is evaluated across multiple models—including ALDI++, ResNet, EfficientNet, and Vision Transformer—demonstrating improved robustness in cross-domain settings.
2. The inclusion of SAM-generated masks makes the method more scalable and practical, reducing reliance on costly manual annotations.

**Weaknesses:**

1. The excessive use of acronyms and abbreviations in the paper makes it very difficult to read. The authors are advised to reorganize and clarify the presentation of this section. In particular, the figures are challenging to interpret.
2. The paper lacks sufficient novelty. The core contribution centers on Equation (2), but the idea of suppressing background and focusing on foreground has already been explored in numerous prior works, and this is not the only possible approach. For instance, could we instead train the model on both the original training set and an augmented version with added background, to prevent the model from over-relying on background cues?

**Questions:**

Please refer to weakness.

---

### Official Review · Reviewer_Yt2i · 2025-10-31

**Soundness:** 2
**Presentation:** 1
**Contribution:** 2
**Rating:** 2
**Confidence:** 4

**Summary:**

This paper addresses the issue in Domain Adaptation for Object Detection (DAOD) where typical detectors use both foreground (FG) and background (BG) features jointly, which harms domain adaptation performance and robustness. To mitigate this, the authors propose a Mask Pooling method that separates FG and BG using masks obtained from either ground truth or SAM. The model then performs feature aggregation on the separated regions. The authors conduct experiments using images synthesized by combining separated FG with random BG, as well as experiments using FG-only images, across multiple datasets and detector architectures, and report performance comparisons.

**Strengths:**

The method is simple and easy to implement, and clearly aims to reduce the blending of FG and BG features, presenting a clear hypothesis and design rationale.

The authors conduct diverse experiments across various datasets and detector models.

**Weaknesses:**

(Major) The performance difference between experiments with no background and those with original background is minimal, which weakens the paper’s central claim that FG–BG feature mixing negatively impacts performance. Moreover, the paper does not provide a direct comparison table between inference with original background and without it. When indirectly comparing Figure 2 and Appendix Table 5, there appears to be no meaningful performance gap—making most of the paper’s main claims unsubstantiated.

The proposed approach replaces the pooling layer, but it is unclear whether this alone effectively mitigates the early FG–BG mixing the paper highlights. Theoretical and experimental justifications for this claim are insufficient.

Although the authors mention it as a limitation, obtaining ground-truth masks in real-world scenarios is impractical, which significantly reduces the applicability of the proposed method.

Since the masking strategy is based on existing ground-truth or SAM-derived masks, the method offers limited novelty.

**Questions:**

See Weaknesses section.

---

### Official Review · Reviewer_jTBd · 2025-11-01

**Soundness:** 1
**Presentation:** 1
**Contribution:** 2
**Rating:** 0
**Confidence:** 4

**Summary:**

The paper investigates a phenomenon termed context bias in object detection methods using causal methods. It proposes a boundary-aware pooling operation with the goal of reducing context bias. It furthermore proposes a DAOD benchmark based on copy-pasted instances on random backgrounds.

**Strengths:**

The paper compares a reasonable set of backbones, detectors, and state-of-the-art DAOD methods. The set of datasets used for benchmarking is reasonable and in line with prior work.

**Weaknesses:**

I find the paper very difficult to follow. For example, section 2 starts with a discussion of causal models of object detection, and arrives at a definition of interventional risk (equation 1), without providing an intuitive explanation or defining variables (for example, how are x and y defined?).

The boundary-aware pooling operation is not particularly novel (see, for example, [1, 2] for similar approaches). That said, if there was a clear and interesting interpretation of this type of pooling from a causal perspective, I think that would be sufficiently novel. Unfortunately, the connection between the causal perspective and the pooling operation is just not sufficiently clearly described.

Some claims seem unfounded (and in fact unachievable), for example: "We also provide a benchmark designed to create an ultimate test for DAOD". It is unclear to me based on the paper what makes the proposed benchmark "ultimate", and I would argue there is no single benchmark for any task that captures all possible relevant aspects.

Some statements seem contradictory, for example: “recent research shows that context bias between FG and background (BG) leads to substantial deterioration of DAOD performance (Son & Kusari, 2024).” vs.
“However, it remains uncertain whether undesirably learned FG-BG association can impact DAOD.”

The paper seems to confuse “models”, “backbones”, and “detectors”, for example in section 3.2 and table 1: ResNets and Efficientnets are backbones, ALDI++ is a detector- / backbone-agnostic domain-adaptation method, and ViTDet is a detector, yet they are all mentioned as “models”. I think the paper would greatly benefit from being more clear about which backbones, detectors, 	and domain adaptation methods were used in which configuration, and where exactly the mask pooling layer fits in.

Overall, I think this paper needs a substantial rework in order to be more approachable and more precise in its language.

[1] https://openaccess.thecvf.com/content_ICCV_2017/papers/Harley_Segmentation-Aware_Convolutional_Networks_ICCV_2017_paper.pdf
[2] https://www.ecva.net/papers/eccv_2018/papers_ECCV/papers/Weiyue_Wang_Depth-aware_CNN_for_ECCV_2018_paper.pdf

**Questions:**

How does boundary-aware pooling mitigate context bias? Wouldn’t backbones still learn to exploit useful signals between foreground and background, with or without boundary-aware pooling?

The caption of Figure 2 says that the figure contains error bars, but I can not seem to find them. Also, what do you mean by "Standard deviation is displayed as error bars but it is trivial"?

---

### Official Review · Reviewer_GsW7 · 2025-11-09

**Soundness:** 2
**Presentation:** 2
**Contribution:** 2
**Rating:** 4
**Confidence:** 3

**Summary:**

This paper tackles context bias in Domain Adaptive Object Detection (DAOD), where models learn spurious correlations between foreground objects and their backgrounds. The authors frame this problem using causal analysis and propose Mask Pooling, a method that explicitly separates foreground and background features during aggregation using masks (either ground truth or SAM-generated). The core contribution is twofold: 1) the Mask Pooling method itself as a causal intervention, and 2) a new evaluation benchmark designed to test model robustness by placing foregrounds onto completely random backgrounds, thereby rigorously measuring the model's reliance on contextual cues.

**Strengths:**

1) The method introduced in the paper is very intuitive and easy to understand.

2) The proposed method achieves strong performance.

**Weaknesses:**

1) The proposed method is overly simple, more like a trick. The core contribution of the paper is in Section 3.1, which is just a pooling operation to separate the foreground and background. This type of processing is quite common in object detection tasks and lacks novelty.

2) The methodology section (Section 3) reads more like a technical report. It describes the method's workflow (or perhaps the experimental workflow) in a very fragmented manner. This makes the paper's quality insufficient to meet the standards for an ICLR paper.

3) Why was it necessary to create a new benchmark?

4) The model's performance improvement also relies on additional computation (e.g., from SAM). The trade-off between computational cost and performance requires further analysis.

**Questions:**

1) A common and more practical method to break FG-BG spurious correlations is using "Copy-Paste" data augmentation, which is similar to the process the authors used to build the random background benchmark. If this Copy-Paste augmentation were used during training, how would its effectiveness compare to Mask Pooling? This seems like a simpler and less computationally expensive alternative.

2) Why was it necessary to create a new benchmark?

---

### Note · Authors · 2025-11-12

**Comment:**

We want to thank the reviewers for their comments. We will take it into consideration and come up with a better version.

**Withdrawal Confirmation:**

I have read and agree with the venue's withdrawal policy on behalf of myself and my co-authors.